# Single-molecule observation of DNA compaction by meiotic protein SYCP3

Johanna L Syrjänen[1†], Iddo Heller[2,3†], Andrea Candelli[2,3], Owen R Davies[4], Erwin J G Peterman[2,3], Gijs J L Wuite[2,3], Luca Pellegrini[1*]

[1]Department of Biochemistry, University of Cambridge, Cambridge, United Kingdom; [2]Department of Physics and Astronomy, Vrije Universiteit Amsterdam, Amsterdam, The Netherlands; [3]LaserLaB, Vrije Universiteit Amsterdam, Amsterdam, The Netherlands; [4]Institute for Cell and Molecular Biosciences, University of Newcastle, Newcastle upon Tyne, United Kingdom

**Abstract** In a previous paper (Syrjänen et al., 2014), we reported the first structural characterisation of a synaptonemal complex (SC) protein, SYCP3, which led us to propose a model for its role in chromosome compaction during meiosis. As a component of the SC lateral element, SYCP3 has a critical role in defining the specific chromosome architecture required for correct meiotic progression. In the model, the reported compaction of chromosomal DNA caused by SYCP3 would result from its ability to bridge distant sites on a DNA molecule with the DNA-binding domains located at each end of its strut-like structure. Here, we describe a single-molecule assay based on optical tweezers, fluorescence microscopy and microfluidics that, in combination with bulk biochemical data, provides direct visual evidence for our proposed mechanism of SYCP3-mediated chromosome organisation.

*For correspondence: lp212@cam.ac.uk

†These authors contributed equally to this work

Competing interests: The authors declare that no competing interests exist.

## Introduction

The synaptonemal complex (SC) is a dynamic proteinaceous ultra-structure that mediates the synapsis of homologous chromosomes pairs during meiotic prophase I (*Cahoon and Hawley, 2016*; *Fraune et al., 2016*; *Yang and Wang, 2009*; *Zickler and Kleckner, 2015*). Morphologically, the SC is composed of three parts: lateral elements, transverse filaments and a central element. Lateral elements are protein scaffolds that extend along the entire length of each chromosome axis. They become linked together by transverse filaments, which connect the chromosomes in a homologous pair to the midline central element, to induce chromosome synapsis (*Schücker et al., 2015*). The tripartite SC ultra-structure provides an essential three-dimensional scaffold for meiotic progression and its absence or defective formation can lead to failure of meiotic progression and infertility (*Bolcun-Filas et al., 2007*, *2009*; *de Vries et al., 2005*; *Hamer et al., 2008*; *Schramm et al., 2011*; *Yang et al., 2006*; *Yuan et al., 2000*).

The lateral element is the first part of the synaptonemal complex to assemble. In most organisms, its formation coincides with the controlled induction of double-strand DNA breaks and subsequent homology searches that bring homologous chromosomes into alignment (*Baudat et al., 2013*). It is at this step that SYCP2 and SYCP3, the major protein components of the lateral element (*Lammers et al., 1994*; *Offenberg et al., 1998*; *Schalk et al., 1998*), assemble on the chromosome axis, in a manner dependent on each other and on the presence of meiosis-specific cohesin (*Fukuda et al., 2014*; *Llano et al., 2012*; *Pelttari et al., 2001*; *Winters et al., 2014*; *Yang et al., 2006*).

Correct SYCP3 function is crucial for fertility. Male mice lacking SYCP3 are infertile as their germ cells undergo apoptosis due to failure in meiotic progression (*Yuan et al., 2002*). In contrast,

females are sub-fertile, with increased aneuploidy leading to fetal deaths during gestation. SYCP3 is regarded as an important architectural component of the lateral element, and its disruption in mice has consequences for the morphology and structural integrity of the meiotic chromosome: in SYCP3-deficient mice, the chromosome axis is twice the length of that in wild-type mice and cohesin disassembles prematurely (*Kouznetsova et al., 2005*; *Yuan et al., 2002*).

The large-scale architecture of the meiotic chromosome consists of a regular array of DNA loops that emanate from a chromosome axis consisting of conjoined sister chromatids (*Zickler and Kleckner, 1999*). Sister chromatid cohesion is enforced at the loop base by meiotic ring-like cohesin complexes (*Rankin, 2015*). In addition to SC components, other factors such as meiotic cohesins, condensins and HORMAD proteins localise to the chromosome axis and contribute to its organization (*Novak et al., 2008*; *Wood et al., 2010*; *Zickler and Kleckner, 2015*). Thus, it is likely that SYCP3's role is to provide local DNA compaction to modulate the existing axis-loop structure of the meiotic chromosome, in the specific manner required for efficient recombination between homologues.

Crystallographic analysis shows that SYCP3 adopts a highly elongated tetrameric assembly of antiparallel α-helices, with flexible termini protruding from the helical core. The N-terminal sequences located at each end of the SYCP3 structure bind double-stranded (ds) DNA, suggesting that SYCP3 can act as a 20 nm-long physical strut to hold distinct regions of DNA together. In addition, SYCP3 can polymerise into regular supra-structures resembling the SC lateral element (*Baier et al., 2007*; *Syrjänen et al., 2014*), in a self-assembly reaction that requires both N- and C-terminal motifs. These findings provide the basis for a possible explanation of SYCP3's role in shaping the architecture of meiotic chromosomes, whereby concurrent DNA bridging and self-association by SYCP3 would explain the observed SYCP3-dependent DNA compaction that is required for full SC assembly and meiotic progression.

To further assess this potential mechanism of SYCP3 function, we developed a single-molecule assay based on optical tweezers and fluorescence microscopy in a microfluidics flow cell. The results of the assay, in combination with bulk biochemical data, provide direct visual evidence for our proposed model of SYCP3-mediated chromosome compaction.

## Results

The crystal structure of SYCP3 showed a tetrameric assembly with N-terminal DNA-binding domains located at both ends of the central helical core (*Figure 1A*). To investigate whether SYCP3 is able to bind simultaneously to distinct DNA molecules as predicted by the structure, we analysed the electrophoretic mobility of SYCP3-DNA complexes using different lengths of dsDNA (*Figure 1B*). To study the mechanism of DNA binding separately from higher-order assembly and alleviate the limited solubility of the full-length protein, we used a version of SYCP3 missing the last six amino acids, SYCP3$_{1-230}$(*Baier et al., 2007*; *Syrjänen et al., 2014*). Hereafter we will refer to this construct simply as SYCP3, unless otherwise indicated.

Reconstitution of an SYCP3 complex with a DNA sample containing equal nucleotide concentrations of dsDNA 32- and 60mer yielded a species of intermediate mobility relative to the slower SYCP3-DNA 32mer and faster SYCP3-DNA 60mer complexes, consistent with the formation of an SYCP3-DNA complex containing both lengths of dsDNA (*Figure 1B*). To provide further evidence of multivalent DNA binding, we showed that SYCP3 can bridge between a fluorescently-labelled dsDNA and a distinct biotin-labelled dsDNA immobilised on streptavidin-coated beads (*Figure 1C*).

To gain mechanistic insight into the mode and dynamics of DNA binding by SYCP3, we took a single-molecule approach using an experimental setup that has been successfully used for the study of protein-DNA interactions (*Brouwer et al., 2016*; *Candelli et al., 2014*; *Heller et al., 2014a*). In this system, the biotinylated ends of a bacteriophage λ DNA molecule are tethered between two streptavidin-coated polystyrene beads that are controlled by focused laser beams acting as optical tweezers. A unique advantage of the use of optical tweezers is the high degree of control over the conformation and tension of the DNA. The experiment is performed in a multi-channel laminar flow cell that allows tethering of DNA to optically-trapped beads and moving the tethered DNA to parallel laminar-flow lanes containing, for example, the fluorescently-labelled components to be analysed (*Figure 2A*). Beam-scanning confocal fluorescence microscopy is then used to image the labelled DNA-bound proteins.

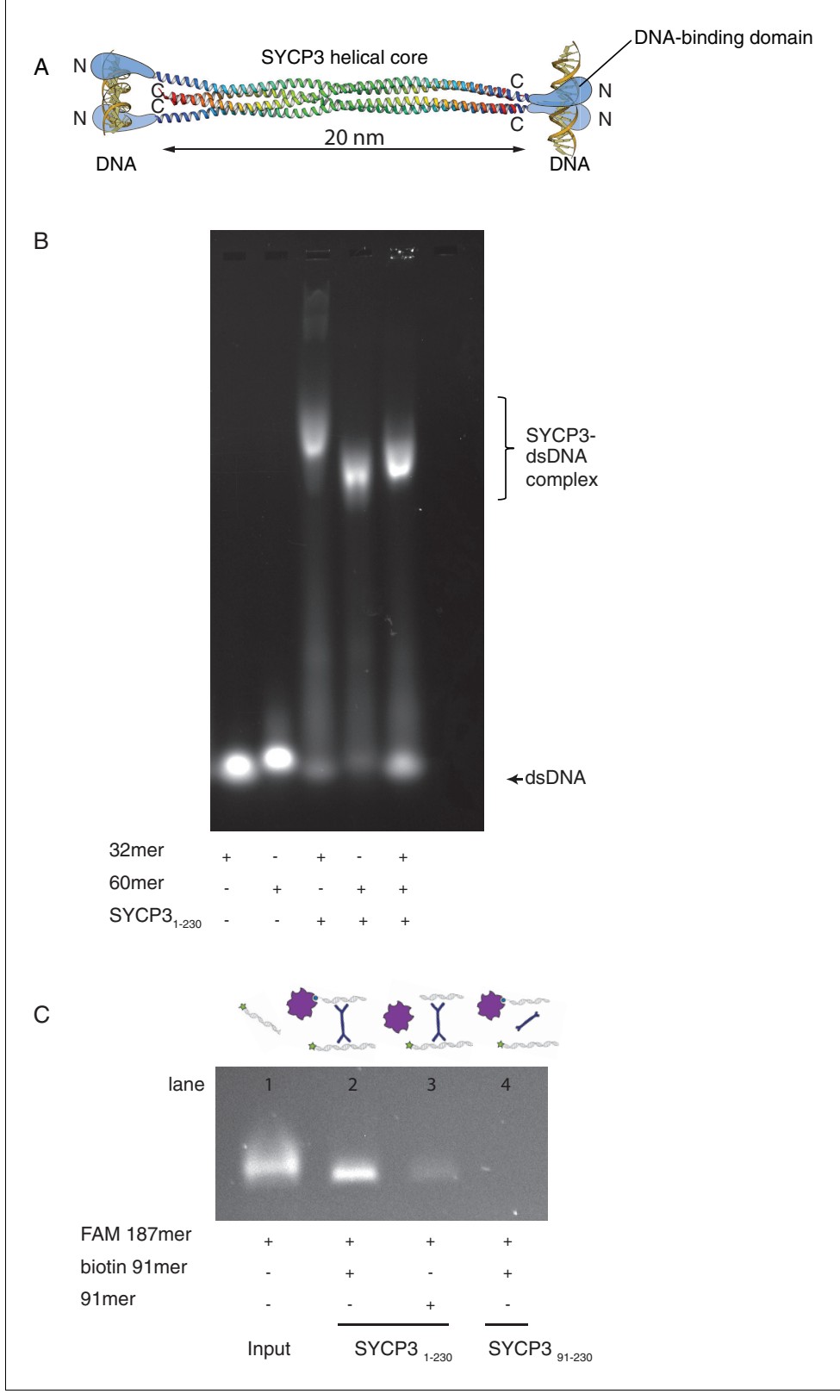

**Figure 1.** SYCP3 can bind simultaneously to two molecules of dsDNA. (**A**) Model of DNA binding by the SYCP3 tetramer. The crystal structure for the helical core of the SYCP3 tetramer is shown in ribbon representation, with each chain rainbow-coloured from the N- (blue) to the C-terminus (red). In the model, pairs of N-terminal SYCP3 regions form two distinct DNA-binding domains at both ends of the SYCP3 strut-like structure. (**B**) Result of the

*Figure 1 continued on next page*

*Figure 1 continued*

electrophoretic mobility shift assay (EMSA) for SYCP3 bound to dsDNA molecules of different lengths. Lanes 1 and 2 show the electrophoretic migration of free dsDNA 32mer and 60mer, respectively. Lane 3 and 4 show the mobility shifts of SYCP3-bound DNA 32mer and 60mer, respectively. Lane 5 shows the mobility shift of a SYCP3-DNA complex in the presence of both DNA 32mer and 60mer species. (C) Result of the pull-down assay, demonstrating DNA bridging by SYCP3. Above each gel lane, drawings illustrate sample content for each pulldown experiment. The 5' FAM-labelled DNA bound to SYCP3 was eluted from the beads in high salt and visualised under UV light after agarose gel electrophoresis (lane 2). Lane 3 shows the background retention level of fluorescently-labelled DNA due non-specific binding of the SYCP3-DNA complex to the streptavidin beads; lane 4 shows that the N-terminal region of SYCP3, known to mediate DNA binding, is necessary for pull-down of the fluorescently-labelled DNA.

We exploited our knowledge of the SYCP3 structure and the absence of cysteines in its sequence to replace solvent-exposed residue L138, located in the helical core and distant from the N-terminal DNA-binding regions, with cysteine (*Figure 2B*). The engineered cysteine was linked to the fluorescent dye Alexa555 using maleimide chemistry (*Figure 2C*), yielding one fluorescent label per helical chain, or four fluorophores per SYCP3 tetramer. Mass spectrometry confirmed that all SYCP3 molecules were labelled with Alexa555, and that the majority of SYCP3 molecules had one label per helical chain (*Figure 2D*).

Binding of fluorescently-labelled SYCP3 to λ DNA in the single-molecule setup was readily observed by confocal fluorescence microscopy. Kymographs, acquired from repeated confocal linescans along the DNA, revealed binding and one-dimensional diffusive behaviour of SYCP3 on dsDNA (*Figure 3A*). As a control, a construct lacking the N-terminal DNA-binding domain, $SYCP3_{91-230}$, did not interact with λ DNA. In 'DNA compaction' experiments, we used the optical tweezers to control the end-to-end distance of the DNA and test the DNA-binding and bridging behaviour of SYCP3 on stretched versus conformationally relaxed DNA. We first incubated the DNA in the protein channel for 30 s, returning subsequently the DNA into the buffer channel, to prevent binding of additional SYCP3 molecules, where it was held in an extended conformation at a force of 5 pN and an end-to-end distance near to 16 μm. Next, we relaxed the conformation of the DNA by moving the beads closer together at constant speed of 0.6 μm/s, reducing the end-to-end distance to 8 μm. The beads were held at reduced distance for a period of 5.5 s, after which the trap lasers were moved apart, extending the DNA molecule again.

The single-molecule analysis highlighted a striking correlation between DNA-binding behaviour of SYCP3 and the distance between the beads tethering the λ DNA (*Figure 3B*). In the initial phase of the compaction experiment, when λ DNA is in an extended conformation, the majority of SYCP3 molecules interacted with DNA in a diffusive manner, displaying a one-dimensional sliding motion along the DNA molecule. Mean-squared displacement analysis of diffusive trajectories revealed a 1D-diffusion constant of $0.16 \pm 0.05$ μm²/s ($N = 8$, see Materials and methods section '1D-Diffusion coefficient analysis'). Intriguingly, when the DNA was temporarily relaxed and then pulled taut again, a drastic alteration in SYCP3's binding behaviour was observed: the majority of diffusing SYCP3 molecules co-localised in clusters that remained static and stable over time.

These findings indicate that SYCP3 has two different modes of binding to DNA. We hypothesise that the diffusive binding mode reflects the interaction with the extended λ DNA of a single DNA-binding domain at one end of the rigid SYCP3 strut, allowing SYCP3 to slide along the DNA molecule (first-binding mode). Upon relaxation of the DNA substrate, DNA-bound SYCP3 would engage concurrently a distinct DNA site with its other DNA-binding domain, in a bridging interaction leading to formation of a DNA loop (second binding-mode). This transient juxtaposition of distinct DNA segments would facilitate bivalent DNA interactions of neighbouring SYCP3 molecules, stabilising the DNA loop and leading to the formation of a static cluster of DNA-bound SYCP3 molecules. Consistent with the presence of SYCP3-dependent loops in the DNA that might resist extension, higher forces were required to extend λ DNA after the compaction step relative to naked DNA (*Figure 4A*). In our experiments, SYCP3-dependent DNA bridges did not exhibit extensive and distinct bond-rupture events (*Figure 4—figure supplement 1*), as previously observed for DNA-bridging proteins H-NS and Alba1 (*Dame et al., 2006*; *Laurens et al., 2012*). The lack of pronounced

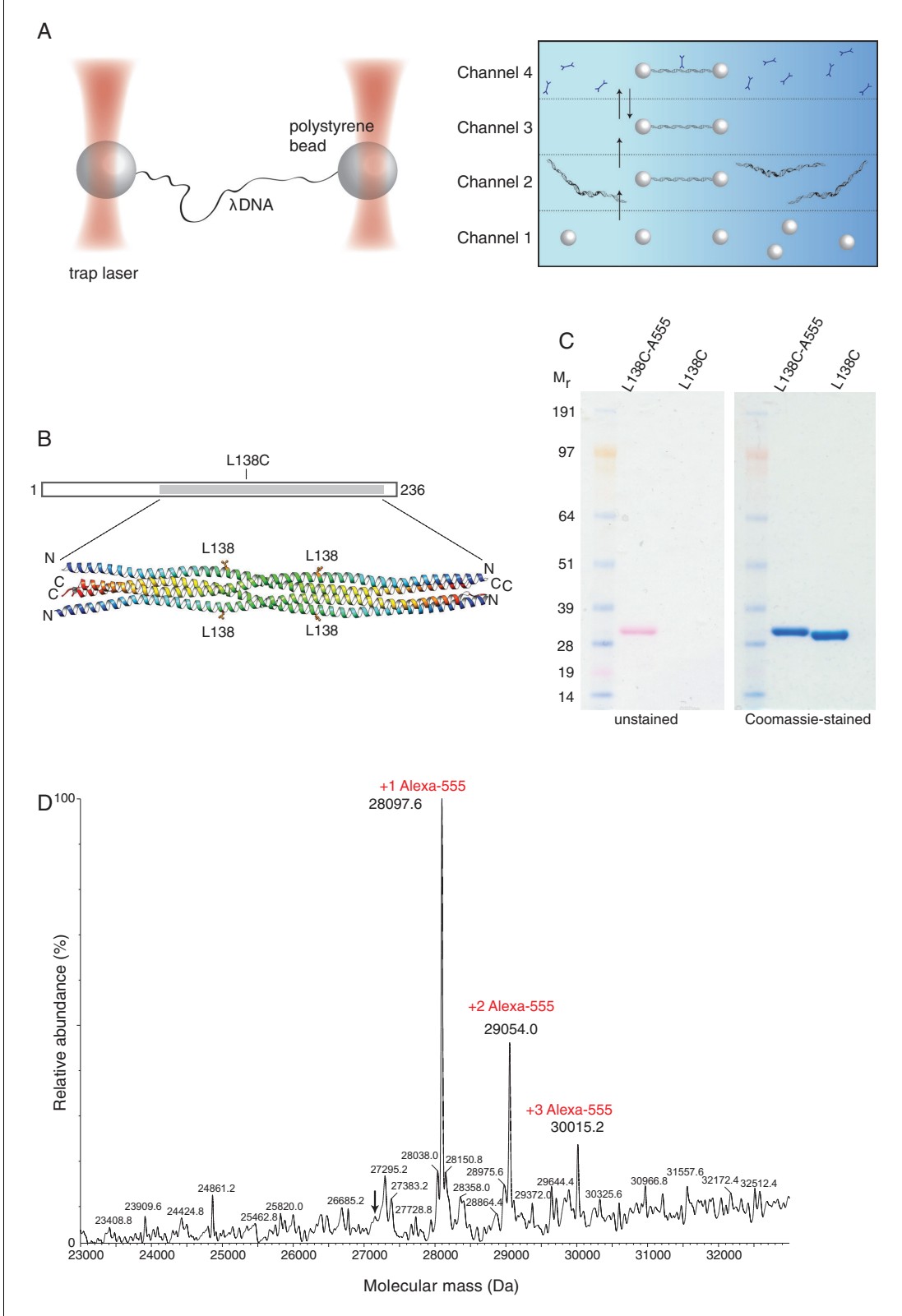

**Figure 2.** Optical tweezers can be used to study the binding of SYCP3 to double-stranded DNA. (**A**) The focused trapping laser beams hold two streptavidin-coated polystyrene beads in place, while the λ DNA is tethered between the beads (right). A four-channel laminar flow cell on a mobile platform was used to perform the single-molecule experiments. The streptavidin-coated polystyrene beads were trapped with the lasers in Channel 1 and biotinylated λ DNA was then captured in Channel 2, by tethering its ends to two beads. The flow cell was subsequently moved to Channel 4, to

*Figure 2 continued on next page*

*Figure 2 continued*

allow fluorescently-labelled SYCP3 to bind λ DNA and, after an incubation time of 30 s, the beads were moved into Channel 3 for visualisation and analysis of DNA binding (left). (**B**) Residue L138 was replaced with cysteine for Alexa555-labelling of SYCP3 via maleimide chemistry. L138 was chosen because it is exposed to solvent and distant from the DNA-binding domains in the SYCP3 structure. (**C**) SDS-PAGE analysis of purified L138C SYCP3$_{1-230}$ both before (L138C sample) and after (L138C-A555 sample) conjugation to Alexa555. The proteins in the gel were visualized in the unstained gel (left) and after staining with Coomassie blue (right). (**D**) Mass-spectrometry analysis shows that the majority of L138C SYCP3$_{1-230}$ contains one Alexa555 label.

rupture events suggests a comparatively high stability of SYCP3-compacted DNA, although further quantitative analysis would require quadruple optical trapping experiments (*Heller et al., 2016*).

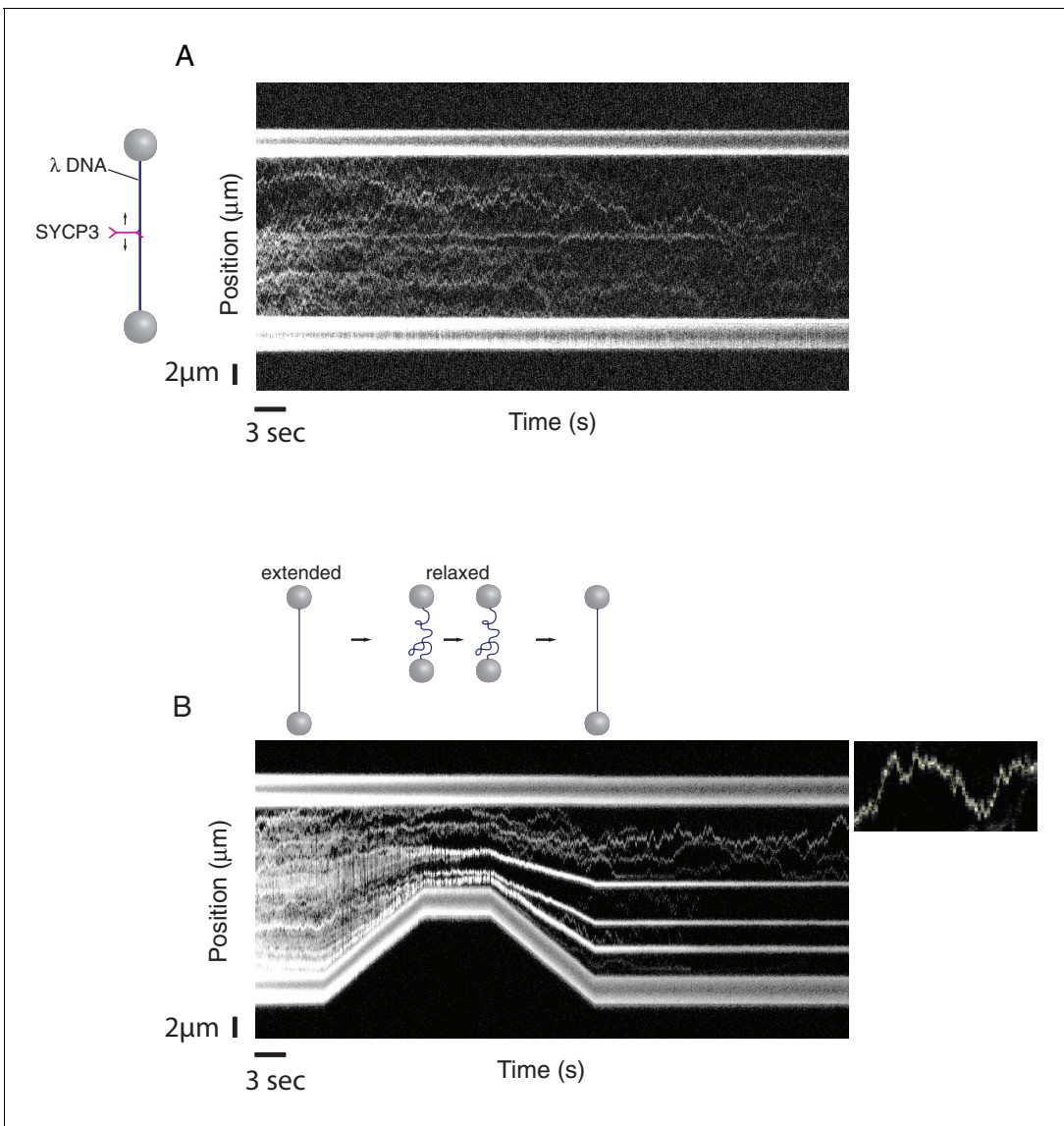

**Figure 3.** A single-molecule DNA-compaction assay shows two modes of SYCP3 binding to dsDNA. Two representative kymographs are shown. (**A**) The kymograph shows the diffusive behaviour observed when SYCP3 was bound to a λ DNA molecule kept in extended conformation by the laser-trapped beads. (**B**) The kymograph in the bottom panel shows clustering of SYCP3 in static aggregates upon relaxation of the λ DNA molecule. The smaller panel shows details of SYCP3's diffusive behavior, over a period of 17.5 s and within a DNA region of 3750 nanometers.

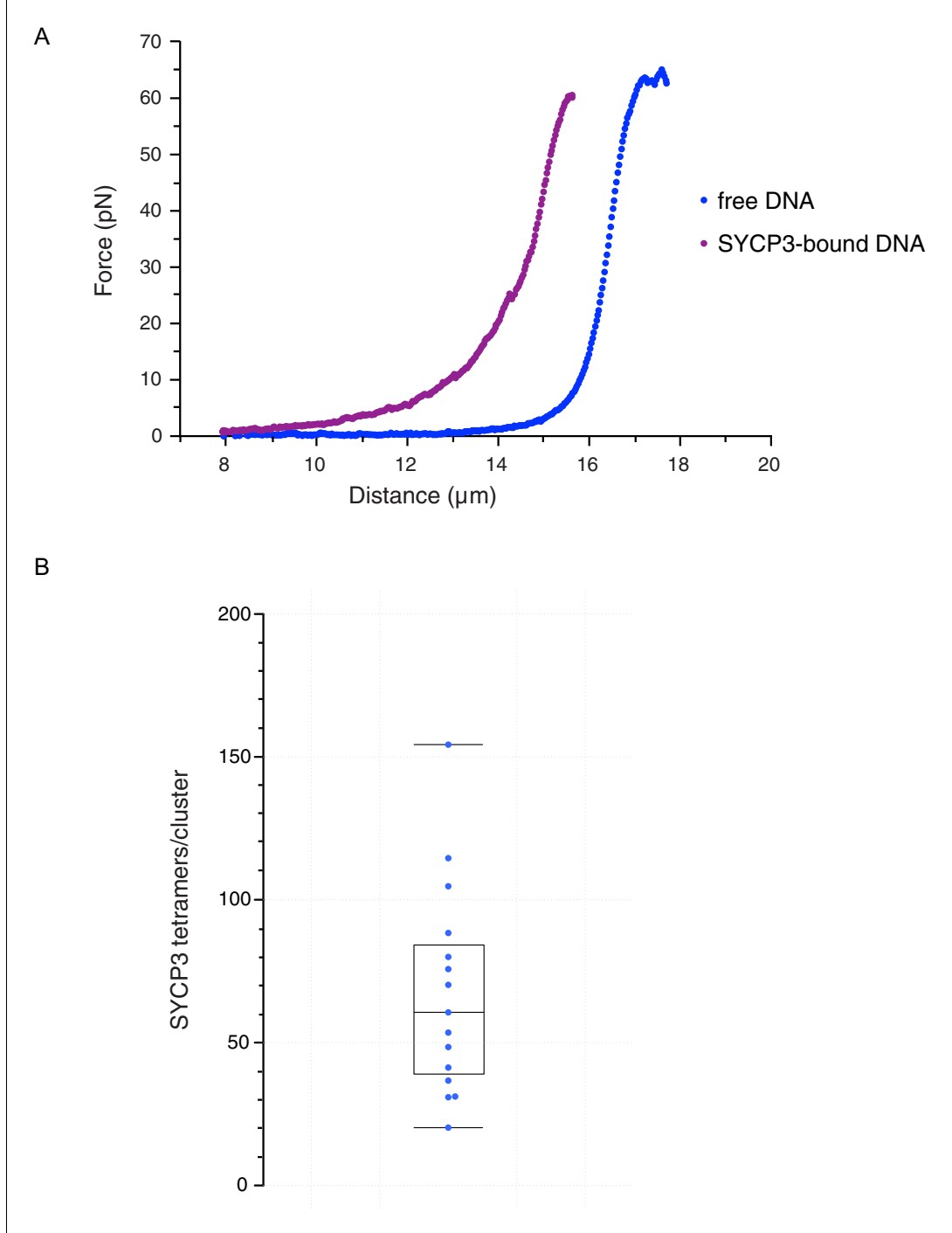

**Figure 4.** Characterisation and analysis of DNA-bound SYCP3 in the DNA-compaction assay. (**A**) Force-distance curves measured for SYCP3-free λ DNA (blue curve), and during re-extension of SYCP3-compacted λ DNA (purple curve). (**B**) Bee-swarm box plot showing the estimated number of SYCP3 tetramers present in the static SYCP3 clusters that form on λ DNA during the compaction assay. The position of the box highlights the interquartile (50%) range. The median, minimum and maximum values are indicated with horizontal lines (line widths are arbitrary).

The following figure supplements are available for figure 4:

**Figure supplement 1.** High stability of SYCP3-compacted DNA.

**Figure supplement 2.** Bleaching rate analysis.

*Figure 4 continued on next page*

*Figure 4 continued*

**Figure supplement 3.** Calculation of the SYCP3 footprint size on DNA based on force - distance curves.
**Figure supplement 4.** DNA-bridging experiments under flow.

The more intense fluorescence signal of the static SYCP3 clusters, relative to the diffuse fluorescence of the mobile SYCP3 molecules before DNA compaction, suggests that they are comprised of numerous SYCP3 tetramers. To obtain a quantitative estimate of the number of SYCP3 tetramers in a cluster, we measured the fluorescence of distinct DNA-bound SYCP3 traces after compaction, by estimating the signal of a single Alexa555 fluorophore and assuming that each SYCP3 molecule bound to DNA as a tetrameric species (see Materials and methods, section 'SYCP3 cluster-size determination'). After correction for bleaching (*Figure 4—figure supplement 2*), we determined that the SYCP3 cluster size was distributed around a mean value of 67 tetramers per cluster, with individual clusters showing a range of 20 to 160 tetramers (*Figure 4B*).

These figures highlight SYCP3's propensity to self-assemble in large DNA-bound aggregates. Combining the quantification of the number of SYCP3 tetramers in a cluster with a measure of the reduction in DNA length caused by SYCP3-dependent compaction yielded an estimate of 2.7 nm ± 1.7 nm (*N* = 3) for the DNA-binding footprint of SYCP3 (see Materials and methods, section 'SYCP3 footprint calculation') (*Figure 4—figure supplement 3*). In a different set of experiments, where flow drag was used to establish micrometer-long sections of DNA molecules bridged by SYCP3 (*Figure 4—figure supplement 4*), we obtained an apparent footprint value of 3.6 ± 0.3 nm (*N* = 13). This footprint value, close to the size of one helical turn of DNA, is in broad agreement with the expected size of the SYCP3-binding site.

## Discussion

Here we have provided direct visual evidence in support of our proposed mechanism of DNA organisation by human SYCP3 (*Syrjänen et al., 2014*). We have shown biochemically that SYCP3 is a multivalent DNA-binding protein and have developed a single-molecule assay to provide experimental evidence consistent with a mechanism of DNA compaction based on non-sequence specific bridging interaction by individual SYCP3 tetramers.

The observed tendency of SYCP3 to accumulate in discrete clusters on dsDNA might form the basis of its ability to compact chromosomal DNA in meiosis. How would transient bridging events account for the formation of stable SYCP3 clusters on dsDNA? Cooperative protein-protein interactions between DNA-bound SYCP3 tetramers might play a role, even in the recombinant SYCP3 protein used for the experiment, which lacked 6 C-terminal residues that are important for self-assembly in the absence of DNA.

Of note, formation of a transient DNA loop by a single bridging event would by itself facilitate further bridging interactions of neighbouring SYCP3 tetramers, even in the absence of cooperative protein-protein interactions (*Figure 5*). Such a mechanism of DNA compaction, driven by the reduced entropic penalty of successive DNA-bridging events, has been predicted by molecular dynamics studies (*Brackley et al., 2013*; *Cheng et al., 2015*). Apparently cooperative DNA-binding behavior of DNA-bridging proteins has been attributed to the induced proximity of distinct protein-bound DNA sites in previous studies (*Dame et al., 2006*). The observed footprint size of SYCP3, corresponding approximately to one helical turn of DNA, is similar to that of the DNA-bridging protein H-NS that was found to bridge two parallel DNA molecules in register with the helical repeat (*Dame et al., 2006*). A preferential DNA-binding orientation correlated to the DNA helix is further supported by the observed value of the 1D-diffusion constant of SYCP3, which is in good agreement with rotation-coupled 1D-sliding along the DNA helix (*Blainey et al., 2009*).

In cells, SYCP3-dependent compaction of the meiotic chromosome takes place in the context of chromatin and in the presence of other essential components of chromosome architecture, such as other SC proteins, meiotic condensins and cohesins, which likely concur to define the extent of SYCP3-dependent DNA compaction during meiosis. Although here we limited ourselves to the study of the binary SYCP3-DNA interaction, our single-molecule setup is well-suited for analysing systems

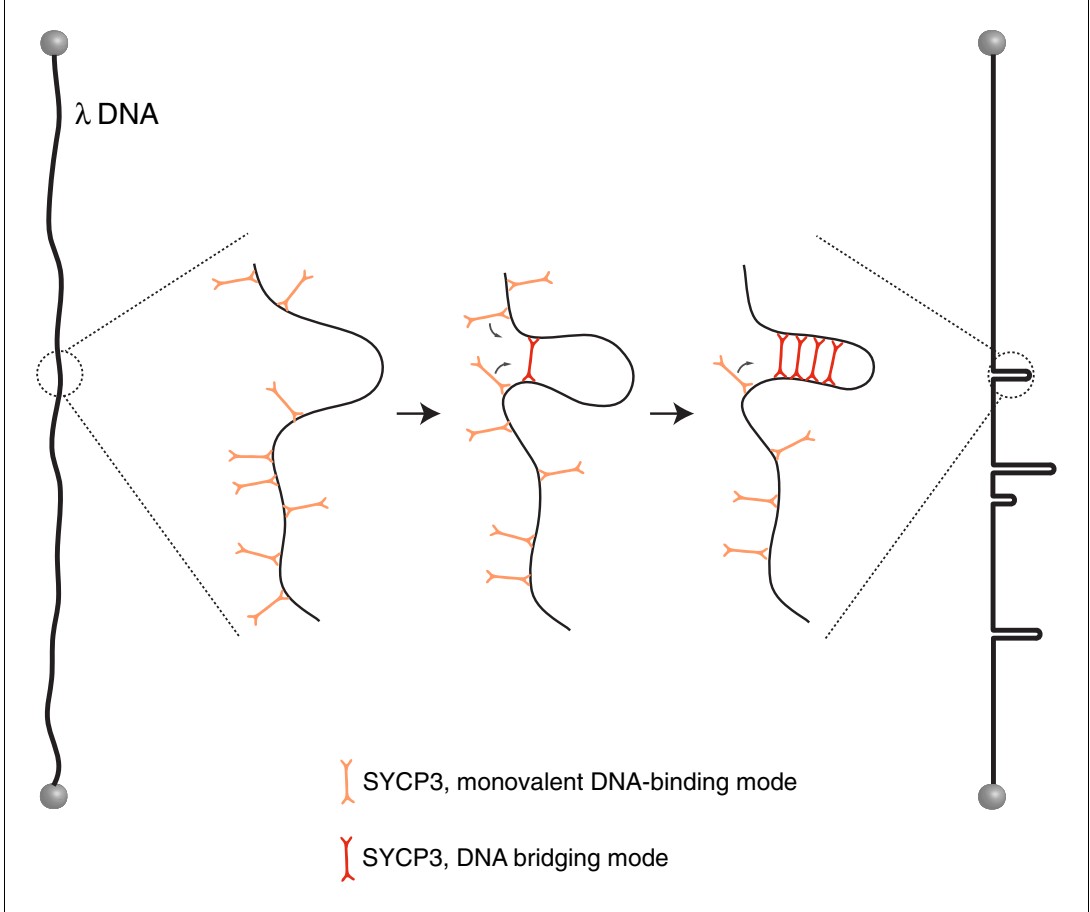

**Figure 5.** Model for the mechanism of λ-DNA compaction by SYCP3, based on its observed behaviour in the single-particle experiments. According to the model, SYCP3 has a diffusive DNA-binding mode, likely mediated by DNA interaction at one end of its strut-like structure, and a DNA-bridging mode, where SYCP3 engages two distinct sites on the DNA with its two DNA-binding domains. DNA bridging by a single SYCP3 molecules can nucleate a DNA loop, which facilitates successive DNA-bridging interactions by other SYCP3 molecules, leading to formation of a stable DNA-bound SYCP3 cluster.

of increasing complexity, including for instance the addition of lateral element protein SYCP2 and core histones. The recent finding that the structure of mitotic chromosomes depends only on a limited number of protein components bodes well for future *in vitro* investigations of meiotic chromosome architecture (*Shintomi et al., 2015*).

## Materials and methods

### Mutagenesis, cloning, expression and purification

The human SYCP3$_{1-230}$ protein was prepared as described (*Syrjänen et al., 2014*). The construct SYCP3$_{91-230}$ was PCR amplified from the full-length SYCP3 construct (*Syrjänen et al., 2014*) and cloned into the bacterial expression plasmid pHAT4 (*Peränen et al., 1996*) using unique NcoI and XhoI sites, for protein expression fused to a N-terminal TEV-cleavable 6xHis-tag. The SYCP3 mutants L138C SYCP3$_{1-230}$ and L138C SYCP3$_{91-230}$ were generated using overlap extension PCR of the full-length SYCP3 construct and cloned as above. All proteins were purified using an initial Ni-NTA capture step, followed by heparin affinity and ion-exchange chromatography, as previously described (*Syrjänen et al., 2014*).

## Electrophoretic mobility shift assay

Double-stranded DNA substrates were generated by annealing together complementary oligonucleotides 60_for, 60_rev and 32_for, 32_rev:

60_for 5'-ATGGTGTGTGTAGGTTAATGTGAGGAGGAGAGGTGAAGAAGGA GGAGAGAAGAAGGAGGC-3'

60_rev 5'-GCCTCCTTCTTCTCTCCTCCTTCTTCACCTCTCCTCCTCACAT TAACCTACACACACCAT-3'

32_for 5'-ATGGTCTGTCTAGGTTACTGTGAGGAGGACGA-3'

32_rev 5'-TCGTCCTCCTCACAGTAACCTAGACAGACCAT-3'

SYCP3$_{1-230}$ was added to a final concentration of 6 μM (per tetramer) to DNA binding buffer (20 mM Tris-HCl pH 8.0, 150 mM KCl) containing 320 μM (per base pair) of linear dsDNA 32mer or 60mer, in a total volume of 10 μl. In the mixed DNA sample, 160 μM of both the 32mer and 60mer were used, so that the total DNA concentration per base pair was 320 μM. DNA samples without added protein were also analysed. 2 μl of 50% glycerol was added to the samples as a loading agent. The electrophoretic mobility of the SYCP3 - DNA complexes were analysed by gel electrophoresis on a 1% (w/v) agarose gel in 0.5xTris-Borate running buffer (Invitrogen) at 4°C at 50 V for 4.5 hr, and visualised by ethidium bromide staining.

## Pull-down assay

The DNA substrates were generated by PCR amplification, to create 5'-FAM 187 bp and 5'-biotin 91 bp DNA molecules. The PCR products were purified using the Thermo Scientific GeneJET PCR purification kit, according to manufacturer's guidelines. 10 μl of streptavidin-linked paramagnetic beads (Dynabeads MyOne Streptavidin T1, ThermoFisher Scientific) were resuspended in 400 μl high-salt buffer (20 mM Tris-HCl pH 8.0, 2 M KCl) and washed three times. The biotin-labelled DNA was immobilised by washing the beads in 40 μl of immobilisation buffer (20 mM Tris pH 8.0, 0.6 mg/ml BSA, 1 M KCl) and adding the biotinylated DNA to a final concentration of 0.1 μM (per molecule). After 30 min incubation with mixing, the supernatant was removed and the beads were washed with 200 μl immobilisation buffer and then with 200 μl of DNA-binding buffer (20 mM Tris-HCl pH 8.0, 150 mM KCl, 0.6 mg/ml BSA). 40 μl of the binding reaction, containing 0.1 μM FAM-labelled DNA and 0.5 μM SYCP3$_{1-230}$ per tetramer in DNA-binding buffer, were added to the beads and the samples incubated with mixing for 15 min, protected from light. The samples were washed twice in 40 μl of DNA binding buffer, and the FAM-labelled DNA was then eluted by adding 40 μl of high-salt buffer and incubating the samples for 30 min, with mixing and protected from light. The eluted DNA was then buffer-exchanged into 10 mM Tris-HCl pH 8.0 using the Thermo Scientific GeneJET PCR purification kit, 2 μl of 50% glycerol was added to 10 μl of sample and the samples were run on a 1% (w/v) agarose gel in 0.5xTris-Borate running buffer (Invitrogen). The result of the pull-down experiment was visualised under UV light.

## Alexa555-labelling of L138C SYCP3$_{1-230}$ and L138C SYCP3$_{91-230}$

The labelling buffer (50 mM Hepes-KOH pH 7.0, 250 mM KCl, 0.5 mM EDTA) was filtered and deoxygenated by bubbling with nitrogen gas for 30 min. The L138C SYCP3$_{1-230}$ protein was buffer-exchanged into the deoxygenated labelling buffer using a NAP-5 column (GE Healthcare). The maleimide derivative of Alexa555 (Thermo Fisher Scientific) in anhydrous DMSO was added drop-wise at a ratio of 5:1 dye molecules to one SYCP3 protomer. The reaction was protected from light and incubated for 2 hr at 4°C with mixing. 10 mM DTT was added to stop the reaction. The labelled protein was then buffer exchanged into 20 mM Tris pH 8.0, 200 mM KCl buffer on a NAP-5 column.

## Single-molecule setup and DNA-compaction assay

A single molecule of linear, double-stranded λ DNA that had been labelled with biotin at both ends was tethered between streptavidin-coated polystyrene beads using optical trapping in a four-channel laminar flow cell, as described previously (*Heller et al., 2013*). 20 mM Tris-HCl pH 8.0, 150 mM KCl was used as the buffer in the flow cell; the protein channel was supplemented with 0.001 mg/ml BSA. Alexa555 L138C SYCP3$_{1-230}$ was used at a concentration of 14.2 nM and Alexa555 L138C SYCP3$_{91-230}$ was used at concentrations of 14.2 nM and 142 nM. Confocal fluorescence microscopy and kymograph recording were performed as previously described (*Heller et al., 2013*)

(*Candelli et al., 2014*). Force-distance data were recorded using custom software written in LabView (National Instruments).

After capture of a single molecule of λ DNA between the optically trapped beads, the flow-cell stage was moved so that the beads were in the waypoint between the buffer and protein channels. The protein channel was opened for 30 s to allow binding of SYCP3 molecules to the DNA, and the flow cell was returned to the buffer channel for visualisation of DNA-bound SYCP3 molecules. Once in the buffer channel, the DNA molecule was held in extended conformation at a force of 5 pN, corresponding to an end-to-end separation near to 16 μm. One of the lasers was then steered towards its neighbour at a speed of 0.6 μm/s so that the beads moved closer to a distance of 8 μm, causing conformational relaxation in the λ DNA. The beads were kept at this separation for approximately 5.5 s and were then moved apart again at 0.6 μm/s to a separation of 16 μm. No flow was applied during force-distance and fluorescence analysis.

## SYCP3 cluster-size determination

Custom software written in MATLAB (MathWorks) was used to analyse the recorded kymographs (available at http://www.nat.vu.nl/~iheller/download.html). The number of photons emitted per Alexa555 fluorophore was determined by fitting a Gaussian distribution to the summed rows of intensities in the bleaching step of a stable SYCP3 cluster. 34 bleaching steps in 10 DNA molecules were analysed in total. The average decrease in amplitude of the Gaussian fit associated with bleaching of a single Alexa555 was 1.5 photons (per confocal scan-line). This implies that the average amplitude of a Gaussian fit for a SYCP3 tetramer is 6.0 photons.

To quantify the number of fluorescent SYCP3 molecules present as stable clusters after DNA compaction, it was necessary to account for bleaching of the fluorophores. The total intensity in photons per pixel over the length of λ DNA ($I_{total}$) was measured at the start of the kymograph (time t = 0) and at a time point when stable clusters had formed (time t), and the ratio $I_{total,t=0}/I_{total,t}$ was used as the bleaching correction factor. The correction makes the assumption that the observed reduction in fluorescence was caused by bleaching, with no dissociation of SYCP3 from the DNA in the experiment. Indeed, under experimental conditions of 6.4 μW excitation intensity, the experiment is dominated by bleaching, with a typical rate of $5 \cdot 10^{-5} s^{-1} \pm 1 \cdot 10^{-5} s^{-1}$ (sd) (*Figure 4—figure supplement 2*). The number of SYCP3 tetramers in a stable cluster was determined by fitting a Gaussian distribution to measure its fluorescence intensity, dividing the amplitude of the Gaussian fit by 6.0 to account for the presence of 4 Alexa555 fluorophores in a tetramer and multiplying by the bleaching correction factor. 15 clusters belonging to four different DNA molecules were analysed in total.

## SYCP3 footprint calculation

To estimate the SYCP3 footprint on DNA, several force-distance curves such as the one shown in *Figure 4A* were analysed. For each DNA molecule, the distance between the beads in the presence or absence of SYCP3 was measured at a force of 25 pN, which is sufficiently high to remove weaker, non-specific interactions of SYCP3 with the DNA and the beads. Thus, the difference in distance values was assumed to be caused solely by the compaction in the DNA resulting from SYCP3's DNA-bridging interactions.

The DNA footprint of each SYCP3 DNA-binding domain was calculated by subtracting from the length difference between free and SYCP3-bound DNA the DNA persistence length (50 nm) multiplied by the number of clusters present in the DNA (to account for the length of DNA at the end of each SYCP3-dependent DNA loop), and dividing the result by the total number of tetramers on the DNA molecule and by 2 (to account for two DNA-binding ends in each SYCP3 tetramer). The calculation was performed on 3 DNA molecules, and gave an average footprint length of 2.7 ± 1.7 nm (sd), corresponding approximately to one turn of B-form dsDNA. The experimental values for DNA length change, number of SYCP3 clusters per λ DNA molecule and total number of SYCP3 tetramers per λ DNA molecule are reported in *Figure 4—figure supplement 3*.

## 1D-Diffusion coefficient analysis

To extract the 1D-diffusion coefficient from our kymograph data, we tracked the diffusive trajectories using tracking software custom-written in Matlab (Mathworks) (available at http://www.nature.

com/nmeth/journal/v5/n8/extref/nmeth.1237-S5.zip). We performed a mean-squared displacement analysis as described in *Heller et al. (2014b)* to obtain the 1D-diffusion constant $D$ for each tracked trajectory. We tracked eight trajectories to find a mean $D$ of 0.16 ± 0.05 μm$^2$/s (sd). The average amplitude of a Gaussian fit to each scan line of the trajectories was 5 ± 2 photons (sd, $N$ = 8). Using the calibrated intensity of 1.5 photons for a single dye (see Materials and methods section 'SYCP3 cluster-size determination'), this corresponds to 3.3 ± 1.6 dyes per diffusing trajectory (sd, $N$ = 8), consistent with the stoichiometry of a single tetramer, with a potential slight underestimation related to photobleaching.

### Rotation-coupled diffusion calculation

We used equation (1) from *Blainey et al. (2009)* to estimate the 1D-diffusion coefficient $D$ assuming rotation-coupled sliding along the DNA helix. A value of $D$ = 0.16 μm$^2$/s can be calculated using, for example, a displacement per helical turn $b$ = 3.4 nm, an average free energy barrier for sliding of $\varepsilon$ = 1.33 $k_B T$, and a protein radius of $R$ = 10 nm, which results in a helical path with radius $R_{OC} = R + R_{DNA}$ = 11 nm. Even though this highly simplified model is based on a spherical protein, unlike the elongated shape of SYCP3, the 1D-diffusion coefficient for pure translational motion along the DNA is ~20 fold larger than that for rotation-coupled sliding.

## Acknowledgements

We would like to thank Joseph Maman for help with Alexa555 labelling of the SYCP3 protein. This work was supported by the European Union's Horizon 2020 research and innovation programme under grant agreement no 654148 Laserlab-Europe, by VICI grants (GJLW and EJGP), a European Research Council starting grant (GJLW) and a H2020 FETopen grant (GJLW and EJGP), by Medical Research Council project grant MR/N000161/1 and Wellcome Trust investigator award 104641/Z/14/Z (LP), and a BBSRC DTP studentship (JLS). ORD is a Sir Henry Dale Fellow jointly funded by the Wellcome Trust and Royal Society (award number 104158/Z/14/Z).

## Additional information

### Funding

| Funder | Grant reference number | Author |
| --- | --- | --- |
| Biotechnology and Biological Sciences Research Council | DTP | Johanna L Syrjänen |
| Wellcome | 104158/Z/14/Z | Owen R Davies |
| Horizon 2020 Framework Programme | 654148 | Erwin J G Peterman<br>Gijs J L Wuite<br>Luca Pellegrini |
| Medical Research Council | MR/N000161/1 | Luca Pellegrini |
| Wellcome | 104641/Z/14/Z | Luca Pellegrini |

The funders had no role in study design, data collection and interpretation, or the decision to submit the work for publication.

### Author contributions

JLS, Conceptualization, Data curation, Formal analysis, Investigation, Methodology, Writing—original draft, Writing—review and editing; IH, Conceptualization, Data curation, Formal analysis, Supervision, Investigation, Methodology, Writing—review and editing; AC, Conceptualization, Supervision, Investigation, Methodology; ORD, Conceptualization, Writing—review and editing; EJGP, GJLW, Supervision, Funding acquisition, Writing—review and editing; LP, Conceptualization, Data curation, Formal analysis, Supervision, Funding acquisition, Writing—original draft, Project administration, Writing—review and editing

**Author ORCIDs**

Owen R Davies, http://orcid.org/0000-0002-3806-5403

Luca Pellegrini, http://orcid.org/0000-0002-9300-497X

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
