## [Decision Letter]

Thank you for submitting your article "Single-molecule observation of DNA compaction by meiotic protein SYCP3" for consideration by *eLife*. Your article has been favorably evaluated by John Kuriyan (Senior Editor) and three reviewers, one of whom is a member of our Board of Reviewing Editors. The following individuals involved in review of your submission have agreed to reveal their identity: Mark Williams and Micah McCauley (together contributed a single review report).

The reviewers have discussed the reviews with one another and the Reviewing Editor has drafted this decision to help you prepare a revised submission

Summary:

This manuscript builds upon a previous *eLife* paper from the same group that provided a structural description of the synaptonemal complex protein SYCP3. In the current manuscript, the authors combine single-molecule optical tweezers, fluorescence spectroscopy and microfluidics to study DNA binding by SYCP3. The ends of this tetrameric protein are shown to bind to and diffuse along DNA. This manuscript shows that multimers of this protein stabilize DNA loops, and suggests that this is done by bridging the sides of DNA loops and stabilizing them. The key quantitative findings of this paper reveal that the binding site size of a single tetramer is 3 nm, and that between 20 and 90 tetramers are found to stabilize each loop.

Essential revisions:

This work is certainly of general interest, both the controls and experiments are well done, and the paper is clearly written. It should be noted that one reviewer raised serious concerns about the physiological relevance of the observations. However, we feel that the demonstration of DNA looping that was hypothesized in the 2014 paper is a sufficient novel advance to be considered for publication as a Research Advance. Before a decision can be made, a number of key shortcomings should be addressed in a revised manuscript (listed below), however. In particular, it will be necessary to provide additional experimental evidence that verifies the loop formation and growth model presented in the manuscript.

As noted by the authors, previous work (Yuan et al. 2002) showed that SYCP3-/- mice have chromosome axes that are roughly 2-fold longer than normal. This is inconsistent with the idea that SYCP3 is a major loop-forming protein – if it were, there would be essentially no axis-loop structure in a SYCP3 mutant, and the chromosomes would be much, much longer. Additionally, a large and growing body of data from cytological, biochemical, and chromosome conformation capture approaches indicates that cohesins bind the base of loops in interphase, mitotic, and meiotic chromosomes. Emerging models also suggest that cohesins actively form the loops, perhaps through an ATP-dependent extrusion activity. SYCP3 is not an ATPase, and simple multivalent DNA binding activity is not sufficient to explain the linear, regular array of loops observed in meiotic chromosomes. In some metazoans, it seems likely that SYCP3 interacts with cohesin-bound DNAs to create regular spacing between the bases of adjacent loops. Alternatively, SYCP3 may impact axis assembly by modulating interactions between cohesins and DNA. The authors should provide more discussion to argue that the evidence in this work implicates SYCP3 as the critical loop-forming protein of the axis, or indeed as a loop-forming protein. The same evidence (that SYCP3 has the ability to simultaneously bind 2 DNA molecules) could be used to argue that the protein mediates synapsis between homologous chromosomes, but it clearly is not in the right place to do this.

This optical trap approach offers great potential to reveal novel and quantitative insights into the organization/condensation of meiotic chromosomes. Optical tweezers might allow the authors to measure relevant biophysical parameters of DNA compaction by SYCP3 such as binding dynamics, typical cluster size or the critical force, as demonstrated by numerous previous studies (e.g. RT Dame, MC Noom, GJL Wuite. Nature, 2006 and many other studies cited in Heller et al. 2014). However, the authors fail to take advantage of their experimental design. In particular, Figure 4 shows one (!) force-extension curve for free and SYCP3-bound DNA, with a comment in the text that "several" such measurements were made. It is not clear from the manuscript how these (this?) force extension curves were acquired. If SYCP3 behaves as suggested by the authors, it should be possible to show stepwise extension of SYCP3-bound DNA using a constant-force regime in the optical trap. The authors should show such stepwise extension data or explain why their experiments did not visualize such behavior.

Other important points to address:

The Methods section refers to Heller et al. 2014, which is a review article that does not include experimental details. Important parameters that should be described in detail include:

1) What is the extension rate in their experiments (this is important, since different extension rates will result in different rupture forces)?

2) How are DNA molecules attached to the beads?

3) Do extension curves in Figure 4 show the same DNA molecule? Since they seem to be relying on absolute lengths for their analysis, are lengths of different DNA molecules identical?

Based on the claim that most SYCP3 molecules are conjugated to a single dye molecule, it should be possible to demonstrate the number of molecules in each "monovalent" binding complex by photobleaching (the fluorescence should bleach in 4 equal, discrete steps). The authors should explain whether the fluorescence data can be used to arrive at conclusions related to stoichiometry.

The footprint calculation and comparison to theory should be more transparent since this was only done for three DNA molecules. What was the length change for each molecule and how many loops did each have? What was the error in the tetramers/cluster of Figure 4? The uncertainty shown is written as "sdom", which we assume means standard deviation of the mean. This is easily confused with standard deviation, so it is preferable to use "sem" for standard error of the mean. If this is what was used, was N taken to be the number of DNA molecules used or the number of clusters?

The authors should better justify their statement that, "SYCP3 is regarded as the main architectural component of the lateral element?" I think many investigators would dispute this idea, and would argue instead that cohesins are the most broadly conserved and thus most essential components.

The authors' model posits that multiple protein complexes merge to form a cluster instead of forming their own mini-bridges. For merging to occur, multiple protein complexes must diffuse to merge. The kymograph in Figure 3 during force reduction suggests that indeed diffusion and merging is happening but no quantification is given.

The beads were held at a short (loop forming) extension of 8 micrometers for 5.5 seconds. If this time were varied, the authors could conclusively determine whether SYCP3 traps a DNA loop of fixed size, which then does not change in time, or if the loop grows continuously with further protein migration along the DNA (as Figure 5 would seem to indicate).

No dissociation of SYCP3 is assumed during the cluster-size determination with photobleaching correction, though this dissociation rate is unknown. Was any effort made to justify this assumption, and to be certain that bleaching was the only reason for the decrease in intensity? One way to resolve this would be to turn the excitation laser off for the duration of an experiment after compaction, and then turn it on again.

In their previous paper, the authors showed that the protein aggregates into large assemblies in the absence of DNA. It is not clear to me how they use the observation of clusters on DNA and interpret these as associated with looping. How do they exclude cluster formation on DNA that is independent on looping? And if aggregation in the absence of DNA occurs, why do they not observe it in the experiment shown in the top panel of Figure 3?

Figure 4 does not seem to support the conclusion in that it does not compare the DNA/protein state before and after a low-tension incubation.

---

## [Author Response]

*Essential revisions:*

*This work is certainly of general interest, both the controls and experiments are well done, and the paper is clearly written. It should be noted that one reviewer raised serious concerns about the physiological relevance of the observations. However, we feel that the demonstration of DNA looping that was hypothesized in the 2014 paper is a sufficient novel advance to be considered for publication as a Research Advance. Before a decision can be made, a number of key shortcomings should be addressed in a revised manuscript (listed below), however. In particular, it will be necessary to provide additional experimental evidence that verifies the loop formation and growth model presented in the manuscript.*

*As noted by the authors, previous work (Yuan et al. 2002) showed that SYCP3-/- mice have chromosome axes that are roughly 2-fold longer than normal. This is inconsistent with the idea that SYCP3 is a major loop-forming protein – if it were, there would be essentially no axis-loop structure in a SYCP3 mutant, and the chromosomes would be much, much longer. Additionally, a large and growing body of data from cytological, biochemical, and chromosome conformation capture approaches indicates that cohesins bind the base of loops in interphase, mitotic, and meiotic chromosomes. Emerging models also suggest that cohesins actively form the loops, perhaps through an ATP-dependent extrusion activity. SYCP3 is not an ATPase, and simple multivalent DNA binding activity is not sufficient to explain the linear, regular array of loops observed in meiotic chromosomes. In some metazoans, it seems likely that SYCP3 interacts with cohesin-bound DNAs to create regular spacing between the bases of adjacent loops. Alternatively, SYCP3 may impact axis assembly by modulating interactions between cohesins and DNA. The authors should provide more discussion to argue that the evidence in this work implicates SYCP3 as the critical loop-forming protein of the axis, or indeed as a loop-forming protein. The same evidence (that SYCP3 has the ability to simultaneously bind 2 DNA molecules) could be used to argue that the protein mediates synapsis between homologous chromosomes, but it clearly is not in the right place to do this.*

We agree that SYCP3’s principal role is probably to provide local compaction of existing loop structures in the meiotic chromosome, rather than being the main driver of large-scale chromosome organization. Indeed, formation of an initial loose looping structure by cohesins and condensins may favour binding and assembly of SYCP3, and thus looping in vitromight be more challenging to achieve. Our findings and our model are fully compatible with this interpretation of SYCP3 function. We have re-written the relevant section of the Introduction (fourth paragraph) to clarify this point.

*This optical trap approach offers great potential to reveal novel and quantitative insights into the organization/condensation of meiotic chromosomes. Optical tweezers might allow the authors to measure relevant biophysical parameters of DNA compaction by SYCP3 such as binding dynamics, typical cluster size or the critical force, as demonstrated by numerous previous studies (e.g. RT Dame, MC Noom, GJL Wuite. Nature, 2006 and many other studies cited in Heller et al. 2014). However, the authors fail to take advantage of their experimental design. In particular, Figure 4 shows one (!) force-extension curve for free and SYCP3-bound DNA, with a comment in the text that "several" such measurements were made. It is not clear from the manuscript how these (this?) force extension curves were acquired. If SYCP3 behaves as suggested by the authors, it should be possible to show stepwise extension of SYCP3-bound DNA using a constant-force regime in the optical trap. The authors should show such stepwise extension data or explain why their experiments did not visualize such behavior.*

We agree with the reviewers that optical tweezers can quantify many biophysical parameters. However, the full range of quantifiable parameters is not experimentally accessible in our current setup, nor is their knowledge essential to support our main conclusion. Here we would like to reiterate that the aim of our study is to provide direct evidence of DNA compaction by SYCP3, rather than investigating in detail the architecture and stability of the resulting protein-DNA structures. Our force- fluorescence data show clearly that upon providing slack to the DNA in presence of SYCP3, the DNA is compacted mechanically in direct correlation to the formation of stable SYCP3 clusters on the DNA.

The reviewers make reference to our previously published experiments (Dame et al., Nature2006), where breakdown of protein-mediated DNA bridges is probed using a specialized instrument that is capable of quadruple optical trapping. Quadruple optical tweezers allow independent manipulation of two DNA molecules, which provides control over the molecular architecture of the DNA bridges. This control is essential to interpret and quantify bridge rupture correctly: in our current dual optical trapping assay (not a quadruple optical trap), there is insufficient control over the molecular architecture, such that forces can be applied over a single terminal bridge in an ‘unzipping’ geometry, or over multiple bridges in a ‘shearing’ geometry, or in a combination of these geometries. This lack of control makes detailed mechanistic interpretation of bridge rupture/rearrangement data unfeasible.

We nevertheless inspected our data for signatures of such bond rupture/bridge rearrangements, and found that the compacted DNA structure appears relatively stable, even at forces approaching the DNA overstretching force of ~65 pN. These observations suggest that SYCP3-DNA bridges appear more stable than previously measured H-NS and Alba1 F60A bridges (Dame et al. Nature 2006 & Laurens et al. Nat Comm2012). We have now added the evidence of long-term stability of the SYCP3-DNA interaction at ~30 pN and ~60 pN to the revised manuscript as a figure supplement (Figure 4—figure supplement 1), and have included a comment in the main text (–Results, seventh paragraph).

*Other important points to address:*

*The Methods section refers to Heller et al. 2014, which is a review article that does not include experimental details. Important parameters that should be described in detail include:*

*1) What is the extension rate in their experiments (this is important, since different extension rates will result in different rupture forces)?*

*2) How are DNA molecules attached to the beads?*

*3) Do extension curves in Figure 4 show the same DNA molecule? Since they seem to be relying on absolute lengths for their analysis, are lengths of different DNA molecules identical?*

We thank the reviewer for pointing out the missing information and the erroneous reference. We have revised the relevant section of the Methods and have updated the reference that describes the details of the experimental procedure, as Heller et al. Nat Meth2013. In answer to the specific questions:

1) The extension rate in the DNA-compaction assay was 0.6 µm/s. This is now mentioned in the text (Results, fifth paragraph).

2) The linear, double-stranded λ DNA molecule is attached to two streptavidin-coated beads by means of biotin labels incorporated at each end of the DNA molecule.

3) The extension curves in Figure 4 refer to the same DNA molecule. All DNA molecules used in the experiments are bacteriophage λ dsDNA, of identical length.

*Based on the claim that most SYCP3 molecules are conjugated to a single dye molecule, it should be possible to demonstrate the number of molecules in each "monovalent" binding complex by photobleaching (the fluorescence should bleach in 4 equal, discrete steps). The authors should explain whether the fluorescence data can be used to arrive at conclusions related to stoichiometry.*

We have indeed observed stepwise bleaching in our fluorescence traces (see Materials and methods, section ‘SYCP3 cluster-size determination’, and bleaching-rate analysis in Figure 4—figure supplement 2). The analysis of bleaching steps was performed on static clusters of multiple SYCP3 tetramers, not on individual tetramers, as our experimental conditions did not allow us to resolve individual DNA-bound tetramers during or right after incubation, due to the high protein density on the DNA. Only after compaction and/or photobleaching we could resolve 1D-diffusive traces that we attribute to individual tetramers. We have added such as trace as an example in Figure 3. The fluorescence intensity of these diffusive traces was 5 +/- 2 photons (N=8), which is consistent with 6.0 photons/tetramer as estimated based on 1.5 photons/dye (see Materials and methods section ‘SYCP3 cluster-size determination’) and a stoichiometry of 4 dyes/tetramer. Photobleaching explains a slightly lower apparent stoichiometry.

*The footprint calculation and comparison to theory should be more transparent since this was only done for three DNA molecules. What was the length change for each molecule and how many loops did each have? What was the error in the tetramers/cluster of Figure 4? The uncertainty shown is written as "sdom", which we assume means standard deviation of the mean. This is easily confused with standard deviation, so it is preferable to use "sem" for standard error of the mean. If this is what was used, was N taken to be the number of DNA molecules used or the number of clusters?*

We have added the requested information relative to length change and loop numbers per DNA molecule in Figure 4—figure supplement 3), and have revised the description of the calculation in the relevant section of the Methods, to make it easier to follow. The error in footprint length is reported as standard error of the mean, as correctly interpreted by the reviewer; we have now changed ‘sdom’ to ‘sem’, as requested. N was taken as the number of DNA molecules analysed in the experiment.

*The authors should better justify their statement that, "SYCP3 is regarded as the main architectural component of the lateral element?" I think many investigators would dispute this idea, and would argue instead that cohesins are the most broadly conserved and thus most essential components.*

We have now modified the relevant section of the Introduction (–fourth paragraph), to provide a more accurate description of SYCP3’s likely role in the organization of the meiotic chromosome structure.

*The authors' model posits that multiple protein complexes merge to form a cluster instead of forming their own mini-bridges. For merging to occur, multiple protein complexes must diffuse to merge. The kymograph in Figure 3 during force reduction suggests that indeed diffusion and merging is happening but no quantification is given.*

We agree with the reviewer that, after putting enough slack onto the DNA molecule, protein complexes can diffuse and merge into a cluster. Since stochastic nucleation and subsequent growth of a cluster causes an increasing fraction of the DNA to be sequestered into protein-stabilized loops, the process terminates when either all of the initial slack in the DNA is removed, or when all DNA-bound SYCP3 ends up in static clusters. The time scale of this termination process is similar to that of the DNA retraction time, in the seconds range.

We have now quantified the 1D-diffusion constant of the mobile SYCP3 molecules (see Materials and methods “1D-Diffusion coefficient analysis” section for details). The 1D-diffusion constant was estimated to be 0.16 µm^2^/s, which is consistent with rotation-coupled diffusion along the helical pitch of the DNA. We now mention the 1D-diffusion constant in the text (Results, sixth paragraph) and comment on it in the Discussion.

*The beads were held at a short (loop forming) extension of 8 micrometers for 5.5 seconds. If this time were varied, the authors could conclusively determine whether SYCP3 traps a DNA loop of fixed size, which then does not change in time, or if the loop grows continuously with further protein migration along the DNA (as Figure 5 would seem to indicate).*

Figure 3 shows that, upon DNA relaxation, SYCP3 concentrates in static clusters and is depleted from flanking DNA regions. This indicates that the protein supply feeding cluster growth has been exhausted and so longer relaxation times would not change cluster size. Under these conditions, we measure SYCP3 clusters of different size. Accordingly, the model presented in Figure 5 shows SYCP3-bound DNA loops of different size, to highlight the stochastic nature of the process of loop nucleation and growth.

Rather than increasing the relaxation time, it might be possible to determine whether an upper limit for the size of SYCP3 clusters exists by repeating the experiment with higher concentrations of SYCP3. However, we believe that this experiment would have only limited physiological relevance, as in the cell nucleus the extent of SYCP3-dependent DNA compaction will be influenced by important protein determinants of chromosome structure such as meiotic cohesin (see for instance, Novak et al., 2008).

*No dissociation of SYCP3 is assumed during the cluster-size determination with photobleaching correction, though this dissociation rate is unknown. Was any effort made to justify this assumption, and to be certain that bleaching was the only reason for the decrease in intensity? One way to resolve this would be to turn the excitation laser off for the duration of an experiment after compaction, and then turn it on again.*

We agree with the referee and have added an analysis of the bleaching rate to the supplementary information (Figure 4—figure supplement 2), in which we varied the excitation laser intensity while monitoring the bleaching rate. Indeed, this analysis confirms that the intensity changes under the experimental conditions is dominated by bleaching.

*In their previous paper, the authors showed that the protein aggregates into large assemblies in the absence of DNA. It is not clear to me how they use the observation of clusters on DNA and interpret these as associated with looping. How do they exclude cluster formation on DNA that is independent on looping? And if aggregation in the absence of DNA occurs, why do they not observe it in the experiment shown in the top panel of Figure 3?*

For our experiments, we used a version of SYCP3 that lacks the last six amino acids (described in the first paragraph of the Results section). This SYCP3 construct retains DNA binding but has lost the ability of DNA-independent aggregation (Syrjanen et al., 2014; Bauer et al., 2007). We are therefore confident that the cluster formation that we observe in our experiments is dependent on DNA binding.

*Figure 4 does not seem to support the conclusion in that it does not compare the DNA/protein state before and after a low-tension incubation.*

We thank the referee for pointing out that our explanation of Figure 4 was not sufficiently clear. The FD curves of Figure 4 were, in fact, acquired before and after a low-tension incubation: the blue curve was acquired on ‘bare’ DNA, in absence of SYCP3, and it represents the ‘before’ situation; the purple curve, on the other hand, was acquired after incubation with SYCP3, and corresponds to the re- extension curve after the low-tension incubation. We have now clarified this in the figure caption and removed the cartoons in the inset of Figure 4 to avoid confusion.